# Detraining Effects of COVID-19 Pandemic on Physical Fitness, Cytokines, C-Reactive Protein and Immunocytes in Men of Various Age Groups

**DOI:** 10.3390/ijerph19031845

**Published:** 2022-02-06

**Authors:** Seung-Jae Heo, Sang-Kyun Park, Yong-Seok Jee

**Affiliations:** 1Department of Physical Education, Chungnam National University, Daejeon 34134, Korea; heosj87@gmail.com; 2Department of Leisure and Marine Sports, Hanseo University, Seosan 31962, Korea

**Keywords:** COVID-19, physical inactivity, immunotoxin, lymphocyte

## Abstract

*Background and Objectives*: Since the start of the COVID-19 pandemic caused by severe acute respiratory syndrome coronavirus II, levels of physical inactivity have become more severe and widespread than ever before. Physical inactivity is known to have a negative effect on the human body, but the extent to which reduced physical fitness has effected immune function before and after the current pandemic has not yet been uncovered. The aim of this study was to investigate the detraining effects of the COVID-19 confinement period on physical fitness, immunocytes, inflammatory cytokines, and proteins in various age groups. The participants of this study included sixty-four male adults who did not exercise during the pandemic, although they had exercised regularly before. *Materials and Methods:* Participants were classified by age group, which included the 20s group (20s’G, *n* = 14), 30s group (30s’G, *n* = 12), 40s group (40s’G, *n* = 12), 50s group (50s’G, *n* = 12), and 60s group (60s’G, *n* = 14). *Results*: Regarding body composition, muscle mass significantly decreased, whereas fat mass, fat percentage, and waist/hip ratio significantly increased in most groups. Cardiopulmonary endurance and strength significantly decreased in all groups, while muscle endurance and flexibility decreased in some groups compared to the pre-COVID-19 pandemic. This study confirmed the immunocytopenia and enhanced inflammation due to physical inactivity during the COVID-19 pandemic, and a greater detrimental decrease mainly after the age of 50. *Conclusion*: This study confirmed a decrease in physical fitness after the start of the COVID-19 pandemic, characterized by an increase in fat mass and a decrease in muscle mass, thereby increasing cytokines and reducing immunocytes in the body. While social distancing is important during the pandemic, maintaining physical fitness should also be a top priority.

## 1. Introduction

Coronavirus disease 2019 (COVID-19) is an infectious disease caused by severe acute respiratory syndrome coronavirus II. People with weakened immune systems and other underlying diseases are especially vulnerable to the COVID-19 outbreak [1]. To minimize person-to-person contact, populations all over the world implemented lockdowns and social distancing measures. Physical activity in public facilities has been restricted to minimize the further spread of infections. In effect, the environment we once knew has changed drastically due to social distancing regulations. It is not an exaggeration to say that people’s immune system is largely dependent on regular physical activity or exercise. It is very important to maintain physical strength and immune function through regular physical activity, especially during this COVID-19 pandemic [2,3].

Immunotoxicity studies show that environmental chemicals can alter the response and function of the immune system and increase immune-mediated diseases [4,5,6,7]. They also reveal that the immune system plays an important role in the pathophysiology of diseases. A pro-inflammatory immune response contributes to tissue and organ damage. Moreover, chronic inflammatory conditions increase C-reactive protein (CRP) concentrations, as well as some cytokines, such as interleukin (IL)-6 and tumor necrosis factor (TNF)-α, leading to immune-related diseases [8]. Similarly, the lack of an appropriate inflammatory immune response contributes to lowered immunity. A thorough understanding of the role of the immune response in the pathophysiology of these diseases is important to identify effective therapies and interventions [9]. Since the immune system is composed of multiple organs and an appropriate immune response involves the interaction of multiple cell types such as dendritic cell, B- and T-cells, and pleiotropic components such as immunoglobulin, CRP, and cytokines, it is important to find out by what factors the human body is affected by the immunotoxin. Immunotoxin effects are commonly categorized into one of four categories: immunosuppression (reduction in the efficacy of the immune system), immunostimulation (general enhanced immune response), hypersensitivity (specific immunostimulatory response mediated by immunoglobulins or T-cells), and autoimmunity. Immunotoxicity refers to any adverse effect on the structure or function of innate and adaptive immunity [9,10,11,12]. A decrease in physical fitness due to a lack of adequate physical activity or exercise is likely to cause immunotoxin disease [10]. There are only a few studies related to the decrease in immunity due to detraining.

These underlying diseases have been reported to increase mortality during the COVID-19 pandemic. There has been much interest regarding the effects of physical fitness on immunocytes, CRP, and cytokines, but research related to reduced physical training or fitness on those variables has not been conducted until now. In addition, studies that focus on various age groups are lacking. Therefore, the aim of this study was to investigate the detraining effects of the COVID-19 pandemic on immunocytes, CRP, and cytokines in various age groups, as well as their effects on physical fitness related to the above variables.

## 2. Materials and Methods

### 2.1. Participants

This study was conducted in an exercise rehabilitation center in Seoul Songdo Hospital and Hanseo University in Korea. Male participants who could not exercise regularly from 9 March 2020 to 28 March 2021 due to COVID-19 social distancing rules were recruited. The average (±SD) period during which the subjects in this study did not use sports facilities and did not participate in regular exercise was 44.85 (±5.65) weeks, although they had exercised for 30–100 min/day, 3–5 days/week for over 6 months prior to the COVID-19 pandemic. All the participants declared no history of having undergone any operation and were without musculoskeletal disorders or cardiovascular problems. Exclusion criteria included having performed any exercise known to affect physical fitness or having undergone any quarantine due to coronavirus infection. The following were also reasons for exclusion: having a history of coronary arterial disease or cerebrovascular disease, an impairment of a primary organ system, severe lung disease, cerebral trauma, uncontrolled hypertension, and cancer. Prior to the study, the participants received detailed explanations regarding all the procedures in this study and were then asked to complete questionnaires. This study was conducted in accordance with the Declaration of Helsinki and was approved by the Institutional Review Board. Written informed consent was obtained before enrollment. Blood samples were collected, and their body composition and physical fitness were assessed.

Eighty-two participants who had undergone physical fitness, cytokine, CRP, and cellular immunity tests for other purposes prior to the COVID-19 pandemic were unable to use public sports facilities due to social distancing after the start of the pandemic. After a roughly one-year hiatus, social distancing was relaxed to some extent, and the subjects who planned to participate in the study a year ago reconvened. As a result, 3 people in the 20s group (20s’G), 4 people in the 30s group (30s’G), 3 people in the 40s group (40s’G), 4 people in the 50s group (50s’G), and 4 people in the 60s group (60s’G) dropped out. Ultimately, this study was initiated because the sample size reached 64 subjects required for the *F*-test. The numbers of participants by age group were 14 in the 20s’G, 12 in the 30s’G, 12 in the 40s’G, 12 in the 50s’G, and 14 in the 60s’G. The complete characteristics of the participants are presented in Table 1.

### 2.2. Measurement Methods

#### 2.2.1. Physical Fitness Measure

The fitness factors were body composition, flexibility, muscle strength, muscle endurance, and cardiorespiratory endurance. The specific measurement method is as follows.

##### Body Composition

Body composition was analyzed by a segmental impedance device that assessed the voltage drop in the upper and lower body (Inbody 770, Biospace Co., Ltd., Seoul, Korea). All participants were asked to remove all metals and anything else that might interfere with the electric stimuli before stepping onto the platform. They were also asked to hold onto the handles and stand still for around 3 min. To minimize error, the participants were asked to avoid food intake for 4 h, alcohol consumption for 48 h, and any kind of exercise for 10 h prior to the test. They were then asked to void 30 min before the test. Before assessing for active fitness factors, each subject performed a standardized warm-up consisting of 5 min of stretching exercises. Body composition variables in this study were adopted as muscle mass, fat mass, body fat percentage, and waist/hip ratio (WHR).

##### Flexibility

Flexibility was assessed using a sit & reach test, which was measured by TKK1859 (Takei Inc., Tokyo, Japan). Cronbach’s α, indicating the reliability of the flexibility test, was 0.725. Participants performed the test with the legs fully extended and the knees relaxed. They were instructed to extend their arms as far as possible and to hold at the furthest point for 2 s. The greatest value after two measurements was recorded.

##### Muscle Strength

Muscle strength was done through a grip strength test using a Smedley dynamometer. It was measured by having the subjects hold the dynamometer so that a right angle was formed by the proximal interphalangeal joint and the width was adjusted accordingly. The arms were placed on the sides in a natural resting position with the dynamometer not touching the body. Both hands were consecutively tested twice, with the maximum value being recorded, and the average of left and right grip strength was analyzed [13]. Cronbach’s α indicating the reliability of the strength test was 0.831.

##### Muscle Endurance

Muscle endurance was done through a sit-up test. Subjects lied down with their backs against the mat with their feet 30 cm apart while an assistant held both ankles down. Both hands were locked behind the head, and the knees were bent at 90°. When the signal was given, the subjects curled their upper body forward so that their elbows touched their knees. Both elbows needed to touch the knees, and the back needed to touch the floor to be counted as one sit-up. The total number of sit-ups completed in 1 min was recorded. Cronbach’s α indicating the reliability of the muscle endurance test was 0.827.

##### Cardiorespiratory Endurance

Maximal oxygen uptake (VO_2_max) for cardiopulmonary endurance was assessed using a graded exercise test. The devices that were used included an electrocardiogram (Q-4500, SunTech Medical, Inc., Morrisville, NC, USA), automatic sphygmomanometer (M-412), gas tester (QMC4200), and treadmill (Q65–90, Quinton, OK, USA). All participants abstained from vigorous physical activity, taking medication 48 h prior to the test, and eating 3 h prior to the test. The electrodes were attached to the chest and the blood pressure cuff was placed on the left brachial artery. The mouthpiece was fixed over the lip and nose area to breath only through the mouth and to the mouthpiece. The Bruce or modified Bruce protocol was used for the participants who continued to walk or run until reaching an all-out level, which was their maximal rating of perceived exertion [14]. The graded exercise test of this study was to investigate exercise capacity. During and after walking or running on the treadmill for as long as possible, the participants were instructed to describe their symptoms as follows: chest pain, shortness of breath, dizziness, and leg pain. The test was terminated if the following symptoms occurred: (a) drop in systolic blood pressure of more than 10 mmHg from baseline, despite an increase in workload, when accompanied by other evidence of ischemia; (b) moderate-to-severe angina; (c) increase in nervous system symptoms; (d) signs of cyanosis; (e) technical difficulties in monitoring electrocardiographic tracings; (f) sustained ventricular tachycardia; (g) ST elevation (>1 mm) in leads without diagnostic Q waves (other than V1 or a VR); and (h) participant’s desire to stop. Cronbach’s α indicating the reliability of the cardiopulmonary endurance test was 0.843.

#### 2.2.2. Immunocytes Measures

Blood samples were collected using BD vacutainer tubes (Becton Dickinson, Franklin Lakes, NJ, USA) to evaluate the immunocytes after fasting for 10 h. After resting for 15 min, 5 mL of blood was collected from the antecubital vein of the subjects with a disposable syringe by a medical laboratory technologist before and after the experiments. Blood samples were left at room temperature for 1 h and centrifuged at 1000 rpm for 15 min for the serum. After that, the immunocytes measures were conducted to evaluate the cellular immune function of the participants, and the distribution and levels of leucocyte, T cells, B cell, and NK cells present in the participants’ peripheral blood were confirmed in this study. The distribution of CD4-positive helper T cells, CD8-positive cytotoxic T cells, and CD56-positive NK cells were measured using flow cytometry [2,3]. In detail, the percentage and absolute cell counts of peripheral blood cell subsets were analyzed as described below: 50 μL of blood was stained with anti-human antibodies against anti-human CD3-Fluorescein isothiocyanate (FITC; Cat No. 555339, BD), CD4-BV510 (Cat No. 562970, BD), CD8-PerCP-Cy5–5 (Cat No. 565310, BD) and CD56-Phycoerythrin (PE; Cat No. 555516, BD) from BD Biosciences (Franklin Lakes, NJ, USA). After incubation for 15 min at room temperature in the dark, the erythrocytes were lysed by adding 450μL of FACS lysing solution to each test tube for another 15 min at room temperature in the dark. They were then analyzed using FACS Canto II (BD Bioscience) and Flowjo software (Treestar, Ashland, OR, USA) and are presented as percentages as shown in Figure 1.

Flow cytometry gating strategy for CD56+ NK cells, helper T cells, and cytotoxic T cells were defined as CD3-CD56+, CD3+CD4+, and CD3+CD8 lymphocytes. Absolute cell counts of lymphocyte subsets were obtained using an automatic hematology analyzer (Sysmex Corp., Kobe, Japan) [3].

#### 2.2.3. Cytokines and CRP Measures

IL-6 and TNF-α concentrations were analyzed using an enzyme-linked immunosorbent assay (ELISA) kit (Cohesion Biosciences, London, UK). The minimum detectable dose of human IL-6 is typically less than 1 pg/mL. The IL-6 ELISA kit allows for detecting and quantifying endogenous levels of natural and recombinant human IL-6 proteins within the range of 3.9–250 pg/mL. TNF-α was allowed to clot in a serum separator tube at room temperature, centrifuged at approximately 1000× *g* for 15 min, and immediately analyzed. The minimum detectable dose of human TNF-α is typically <7 pg/mL. The TNF-α ELISA Kit allows for the detection and quantification of endogenous levels of natural and/or recombinant human TNF-α proteins within the range of 15.6–1000 pg/mL. The CRP test was carried out using the immunoturbidity-based CardioPhase hsCRP (Dade Behring, Marburg, Germany) reagent mounted on the BNII (Dade Behring, Marburg, Germany) equipment. The full-scale CRP test using the immunoturbidity method was performed using Nanopia CRP (N-CRP; Sekisui, Tokyo, Japan) and IATRO CRP-EX (I-CRP; Iatron, Tokyo, Japan). CRP reagents with PureAuto S CRP-SS type (P-CRP; Sekisui) were installed on a Hitachi 7600–110 automated analyzer (Hitachi High-Technologies Co., Tokyo, Japan), and the dilution factor of the equipment was not changed. The minimum detectable dose of human CRP is usually <10 pg/mL. The CRP ELISA kit allows for the detection and quantification of endogenous levels of natural and recombinant human CRP proteins within the range of 15.6–1000 pg/mL. Afterward, materials were removed by washing, and a biotinylated polyclonal antibody specific for biomarkers was added to each well. Unbound antibody-enzyme for biomarkers was removed by washing, and horseradish peroxidase was added to each well [15].

### 2.3. Data Process and Statistical Analyses

G*Power 3.1.9.7 was used to calculate the sample with an effect size of 0.40, a significance level of 0.05, and a power of 0.80 required for the ANCOVA test. Effect size (η^2^) was calculated according to Cohen’s *d,* which is equal to the mean difference of the groups divided by the pooled SD [16]. All data were reported as mean ± standard deviation and carried out using IBM^®^ SPSS^®^ Statistics software (version 25.0. IBM Corporation; Armonk, NY, USA). The distribution of all data was checked using the Shapiro–Wilk test. Age, height, and body weight measured using the Kruskal–Wallis test prior to this study showed significant differences among the groups. Thus, using these three variables as covariates, MANCOVA was performed, and the difference among the 5 groups after treatment was verified. A Wilcoxon rank test was conducted to investigate the differences between pre- and post- values in each group. At last, delta (Δ) % was calculated for each period for detailed data analysis. For all analyses, the significance level was set at *p* ≤ 0.05.

## 3. Results

### 3.1. Analysis of Demographic Variables and Body Composition

As shown in Table 1, although age, height, and body weight were significantly different among the groups, the percent fat (*p* = 0.869) and waist/hip ratio (WHR, *p* = 0.114) of the five groups were not significantly different. As shown in Table 2, muscle mass was highest in 30s’G and 40s’G, while it was lowest in 60s’G before the COVID-19 pandemic. This trend was similar during the pandemic. Characteristically, muscle mass significantly decreased in all groups except 20s’G during the COVID-19 pandemic. Meanwhile, fat mass, percent fat, and WHR before and during the COVID-19 pandemic did not differ significantly between groups. These variables showed increased results in each group as well.

As shown in Figure 2, after detraining due to COVID-19, the muscle mass in 20s’G decreased by about 8%, 30s’G by 11%, 40s’G by 10%, 50s’G by 13%, and 60s’G by 24%. After detraining, fat mass increased by about 24% in 20s’G, 14% in 30s’G, 30% in 40s’G, 25% in 50s’G, and 34% in 60s’G. Similar to fat mass, fat percentage increased by about 18% in 20s’G, 12% in 30s’G, 27% in 40s’G, 30% in 50s’G, and 33% in 60s’G. WHR showed a steady rise with increasing age. That is, in the case of 20s’G, WHR increased by about 7%, 30s’G by 10%, 40s’G by 9%, 50s’G by 11%, and 60s’G by 15%.

### 3.2. Comparisons of Physical Fitness

As shown in Table 3, all physical fitness factors were not significantly different among the groups before and during the COVID-19 pandemic. Specifically, VO_2_max, which was used to measure cardiopulmonary endurance, and grip strength used to assess muscle strength, were significantly reduced in all the groups during the COVID-19 pandemic. Sit-ups used to assess muscle endurance significantly decreased in all the groups except for 40s’G. Sit & reach, which was used to test flexibility index, significantly decreased except for 30s’G and 40s’G.

As shown in Figure 3, after detraining due to COVID-19 confinement, the cardiopulmonary endurance in 20s’G decreased by about 6%, 30s’G by 13%, 40s’G by 9%, 50s’G by 17%, and 60s’G by 18%. After detraining, strength decreased by about 15% in 20s’G, 9% in 30s’G, 10% in 40s’G, 27% in 50s’G, and 20% in 60s’G. Similar to strength, muscle endurance decreased by about 9% in 20s’G, 12% in 30s’G, 11% in 40s’G, 17% in 50s’G, and 22% in 60s’G. Flexibility also decreased after detraining. In 20s’G, it reduced by about 45%, 30s’G by 17%, 40s’G by 18%, 50s’G by 40%, and 60s’G by 42%.

### 3.3. Comparison Results of Immunocytes

As shown in Table 4, leucocytes and lymphocytes were not significantly different among the groups before and during the COVID-19 pandemic, except for NK cells. Specifically, only NK cells showed no significant difference between groups before the pandemic; however, there were significant differences between groups during the pandemic. After the start of the pandemic, CD3, CD4, and CD8 of the T cells showed a tendency to decrease in all groups, although there was no significant difference.

### 3.4. Comparison Results of Cytokines

As shown in Table 5, IL-6 was highest in 50s’G, while it was lowest in 20s’G before the COVID-19 pandemic, whereas this trend was not significantly different among the groups during the pandemic. IL-6 significantly increased in all groups except for 50s’G during the pandemic. TNF-α appeared similar to IL-6 in terms of time-to-time changes and differences between groups. There was a significant difference in the aspect of CRP levels before and during the pandemic. Specifically, CRP was highest in 60s’G, while it was lowest in 20s’G before and during the pandemic, although CRP significantly increased in all groups during the pandemic.

As shown in Figure 4, after detraining due to COVID-19, the IL-6 in 20s’G increased by about 18%, 30s’G by 21%, 40s’G by 22%, 50s’G by 16%, and 60s’G by 47%. After detraining, TNF-α increased by about 37% in 20s’G, 39% in 30s’G, 38% in 40s’G, 9% in 50s’G, and 25% in 60s’G. CRP also increased by about 31% in 20s’G, 41% in 30s’G, 32% in 40s’G, 31% in 50s’G, and 17% in 60s’G. Meanwhile, CD56 decreased by about 7% in 20s’G, 10% in 30s’G, 7% in 40s’G, 14% in 50s’G, and 31% in 60s’G. CD3 decreased by about 13% in 20s’G, 22% in 30s’G, 21% in 40s’G, 26% in 50s’G, and 39% in 60s’G. CD4 decreased by about 11% in 20s’G, 13% in 30s’G, 16% in 40s’G, 19% in 50s’G, and 15% in 60s’G. At last, CD8 decreased by about 23% in 20s’G, 28% in 30s’G, 32% in 40s’G, 8% in 50s’G, and 13% in 60s’G.

## 4. Discussion

The main result of this study was that the COVID-19 period negatively affected all physical fitness factors. Although a significant difference in muscle endurance was not shown in 40s’G, and flexibility did not significantly decrease in 30s’G and 40s’G, cardiopulmonary endurance and muscle strength significantly decreased in all groups during the COVID-19 pandemic. Such deterioration in physical fitness factors negatively affected body composition, immunocytes, cytokines, and inflammatory protein. The results of this study were similar to the results of other studies in that the physical fitness improved when exercise was performed. In contrast, the improved levels of physical fitness returned to baseline levels or worsened when training was stopped.

Cardiorespiratory endurance is the most basic element of health-related fitness. Muscle strength, endurance, and flexibility are also important for health, but these factors are most effective when the cardiopulmonary system is strong. The role of removing waste products by supplying nutrients and oxygen to body tissues depends on the ability of the lungs, heart, and vessels. All human beings can respond or adapt to physical needs only when they are in good physical condition. Regular physical training increases the efficiency of the heart, from which a large amount of blood is drawn per heart rate, and increases the VO_2_max. On the other hand, when training is discontinued, which occurred for many people due to social distancing measures, the efficiency of the cardiopulmonary system decreases, and VO_2_max is reduced. A study by Costill et al. [17] showed that aerobic capacity decreases by 10–15% when endurance athletes stop training for a week, while Fox et al. [18] indicated a period of 4 to 8 weeks to lose most of the benefits obtained from physical training, including changes in muscle enzyme activity and function. According to Saltin and Rowell [19], the maximal cardiac output during cardiopulmonary exercise begins to decrease between 5–12 days of discontinuation, and the decrease in maximum cardiac output reduces blood flow that carries oxygen to the muscle fibers. As in previous studies, when estimating the delta% of VO_2_max after the experiments, the VO_2_max in 20s’G, 30s’G, 40s’G, 50s’G, and 60s’G decreased by −5.6%, −13.3%, −8.9%, −16.9%, and −17.8%, respectively. When averaging the delta% for all age groups, it appeared to have decreased by over −13%, as shown in Figure 3.

Muscle strength is an important factor for health, as well as movements for all sports activities. Physical fitness training expands muscle tissue to increase relative adaptive capacity, while exercise capacity decreases when muscle strength weakens. According to Graves et al. [20], even if training frequency is reduced from 1 to 2 days per week, muscle strength can be maintained. It has been reported that, even when exercise is discontinued, muscle strength is generally better maintained than other fitness factors. However, Hakkinen and Komi [21] reported that even elite weightlifters who trained for 24 weeks had a 10–30% decrease in muscle strength by 4 weeks when they stopped training for 22 weeks. This study also showed that muscle strength in 20s’G, 30s’G, 40s’G, 50s’G, and 60s’G decreased by −14.8%, −9.3%, −9.9%, −26.8%, and −19.6%, with an average decrease of−16.1% compared with last year. Similar to the decrease in muscle strength due to training cessation, there is a significant decrease in muscular endurance as well. According to Fleck [22], the decrease in muscle endurance is a decrease in the ability of the skeletal muscles to use oxygen from the bloodstream, which is closely related to the deterioration of capillaries, blood flow, and oxidative capacity of the skeletal muscles. Houston et al. [23] demonstrated that the capillary density of skeletal muscles is significantly reduced after 15 days of training for endurance athletes. The results of these previous studies are consistent with the results of this study. It was found that the average decrease in muscle endurance (−14.4%) was due to the closure of sports facilities during the COVID-19 pandemic. Among the health-related fitness factors, flexibility indicates the range of motion of a joint, which can be diminished by a lack of exercise. A reduction in flexibility not only decreases the elasticity of ligaments and tendons due to a shortened range of joint movement, but also lowers muscle strength [17]. Decreased flexibility due to lack of exercise or training can lead to incorrect posture in the main joints of the human body, resulting in injury. According to the results of this study, flexibility was significantly reduced in men 20 years old and over 50 years old who stopped physical activity during the pandemic.

As is common knowledge, we know that regular exercise leads to desirable changes in body composition. In other word, a healthy physical fitness is maintained through repeated contraction and relaxation of muscles, but a decrease in sustained activity has several negative consequences. Ogden et al. [24] reported that regular exercise is crucial for losing body weight and is a predominant component in lifestyle modifications. Some researchers reported that regular exercise could decrease subcutaneous adipose tissue deposition and reduce visceral adiposity [25,26,27]. As with other fitness factors, the body composition is also negatively changed by the cessation of regular training. Even looking at the result of this study, muscle mass significantly decreased in all ages, whereas fat mass, fat percentage, and WHR during the pandemic significantly increased compared to body composition before the pandemic. Specifically, muscle mass markedly decreased in the 60′sG (−24.5%), while fat mass significantly increased, as shown in Figure 2. The skeletal muscle system adapts to prolonged physical inactivity, decreasing the size of the muscle fiber, in addition to the loss of muscle function and quality [9].

Excluding body composition, several researchers indicated that the inflammatory cytokines and proteins changed negatively due to decreased muscle mass and increased fat mass [28,29]. Cytokines are immunomodulatory transport proteins secreted by immunocytes. As a substance that facilitates cell-to-cell communication, it mediates the interaction between cells necessary for an individual’s cells and tissues to exhibit an organic or synthetic response to various stimuli from the outside. In addition, as a water-soluble extracellular protein or glycoprotein, it is shown not only for intracellular mediators and aggregates, but also for self-defense against inflammation, cell growth, differentiation, apoptosis, angiogenesis, and restoration and development of homeostasis [30]. Cytokines are not always present in serum, and their production is usually transient. In addition, cytokines have various characteristics as proteins of various terminations. These are produced by various immunocytes and affect the activation, growth, and differentiation of the cells. Another characteristic of cytokines is that they are small-molecule proteins that help regulate and interact between cells, including immune responses, and increase when muscle cells are injured during exercise, which sequentially releases TNF-α and IL-6. This study also showed that inflammatory cytokines increased even when exercise was stopped, offering a mechanism of reduced cellular immune function, increasing susceptibility to infection. Recently, Sharif et al. [31] reported that physical activity leads to a significant increase in T cells, decreased immunoglobulin secretion, and produces a shift in the helper T (Th)1/Th2 balance to a decreased Th1 cell production. Moreover, they showed that physical activity promoted the release of IL-6 from skeletal muscles, which functions as a myokine and has been shown to induce an anti-inflammatory response. However, changes in cellular immune function are reversed when exercise is not performed. In other words, the inflammatory index in the body increases, whereas the function of immunocytes decreases.

Early reports on COVID-19 infections described an associated upsurge of pro-inflammatory cytokines [32]. In addition to IL-6, other inflammatory protein such as CRP, serum ferritin and coagulation index, and D-dimer were elevated in patients with COVID-19 and diabetes compared to those without diabetes, indicating that patients with comorbidities are more prone to an inflammatory response [33]. A plasma CRP is an annular pentameric protein whose circulating concentrations rise in response to inflammation. It is an acute-phase protein of hepatic origin that increases following IL-6 secretion by macrophages and T cells. Chronic inflammation has been considered to be an important contributing factor for sarcopenia [34,35]. Elevated levels of several circulating cytokines have been shown to correlate with the occurrence of sarcopenia [36]. IL-6, TNF-α, and CRP levels have been reported to be significantly upregulated in patients with sarcopenia. This study also showed that IL-6 (46.6%), TNF-α (24.6%), and CRP (17.2%) significantly increased in male subjects in their 60s when a decrease in muscle mass (−24.5%) was evident during the pandemic, as shown in Table 5. In other words, this response is lost in older human skeletal muscles suggesting that aging impairs the response of skeletal muscles to exercise [37]. TNF-α is a pro-inflammatory cytokine present in skeletal muscle microenvironments. It has been shown to promote muscle regeneration in young mice by improving muscle stem cell proliferation. TNF-α is also increased in aged muscle [38]. The increased amount of TNF-α further switches on TNF-α signaling. Indeed, the muscle atrophy in type II muscle fibers upon aging is correlated with a higher activity of caspases [39].

To protect ourselves during the ongoing COVID-19 pandemic, it is most important to keep the immune function from declining due to poor physical fitness. Regular exercise is essential to maintain immune function in a healthy state and to strengthen immune function. Many studies have been conducted to investigate the relationship between exercise and immunity. Exercise has been identified as a behavioral factor that can boost immune function in some settings and, therefore, potentially serve as an adjuvant for immune responses [40,41,42,43,44]. There is a positive association between exercise and immune response to vaccination, particularly in populations at risk for immune dysfunction, such as older adults. Studies investigating the effects of regular exercise on the immune response to vaccination have consistently found that it can be enhanced through moderate exercise training in older adult populations. Several studies showed greater immune responses to vaccination in habitually physically active adults who received exercise training [45,46,47,48,49]. Cells related to immune function are affected by exercise. In particular, it has been shown that exercise affects T cells and various types of immune-related proteins. Lin et al. [50] reported that T cells increased even after a short bout of exercise. They also reported that exercising at 60–70% of maximal oxygen consumption for 30–60 min per day, 5 days per week for 10 weeks, decreased the mitogenic activity of spleen lymphocytes to concanavalin and staphylococcal enterotoxin, increased the proliferative response to lipopolysaccharide, and reduced IL-2 production in the trained group. Peake et al. [51] indicated that exercise induction of a pro-inflammatory environment in the muscles, especially in the case of muscle-damaging exercise, may result in increased lymphocyte homing to the site of vaccine administration and enhanced antigen uptake and processing, making the initial phase of the immune response more efficient. Evidence observed in previous studies suggests that yoga can improve the immune system. There was an increased level of CD4 and CD8 lymphocytes and NK cells after practicing yoga compared to the control group [52]. Rajbhoj et al. [53] reported beneficial changes in the immune system, which led to a decrease in pro-inflammatory cytokines and an increase in anti-inflammatory cytokines in those who practiced yoga. The group that performed yoga had higher values of CD 4 and CD8 lymphocytes and NK cells than the control group. On the other hand, a lack of exercise and sarcopenia negatively affected cytokines, as well as immunocytes. In other words, some immunotoxicity can occur. Countless studies have documented how abnormal responses from the immune system and the triggering of local inflammation in key cells and tissues are the first causes in the development of chronic diseases [54]. This study found that the ratio and function of NK cells responsible for innate immunity and T cells (CD3, CD4, and CD8) responsible for acquired immunity decreased during the COVID-19 pandemic. Similar to previous studies, these results showed a noticeable decrease mainly in subjects in their 60s. Indeed, regular physical activity or exercise induces a cascade of benefits that strengthen the immune system via an ultimately anti-inflammatory pattern within organ crosstalk [55]. Regular exercise or physical activity helps improve body composition and immune function, which leads to better health. It would be very beneficial to find an appropriate exercise intensity, duration, and frequency that improves immune function, especially in the midst of this pandemic. Although practicing social distancing is essential, maintaining physical fitness should also be considered a top priority. It would be beneficial for further studies to investigate the detraining effects of COVID-19 confinement on a greater number of participants with diverse demographic backgrounds and multiple immune-related cell tests.

## 5. Conclusions

Combining the results of this study, it was confirmed that the COVID-19 pandemic period caused a decrease in physical fitness, in particular, an increase in fat mass and a decrease in muscle mass, thereby causing immunocytopenia and hypercytokinemia in the body. These results can lead to a decrease in health and immunity due to detraining, which can lead to greater susceptibility of infection. Therefore, a strategy that promotes regular physical exercise is needed, even before COVID-19 vaccines and therapies are developed and distributed.

## Figures and Tables

**Figure 1 ijerph-19-01845-f001:**
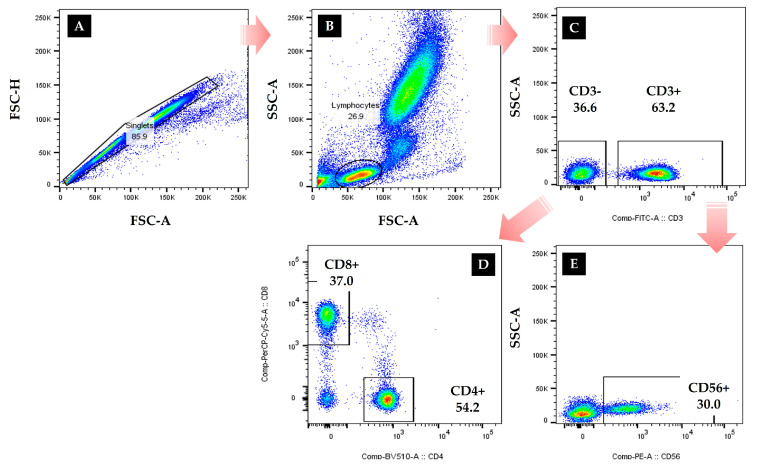
Flow cytometry. Singlet cells were gated by area and height of forward scatter. A front scattered light (FSC)-A and side scattered light (SSC)-A plot was used to identify nucleated cells. (A) Single cells were gated, and doublets were excluded. (B) Lymphocyte gate was set based on the size and granularity of the cells. (C) CD3- and CD3+ were excluded. (D) CD8+ and CD4+ were defined. (E) CD56+ was identified.

**Figure 2 ijerph-19-01845-f002:**
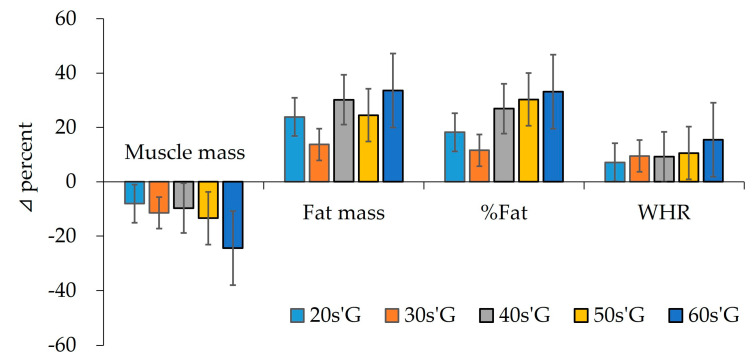
Delta % in body composition factors among groups after detraining.

**Figure 3 ijerph-19-01845-f003:**
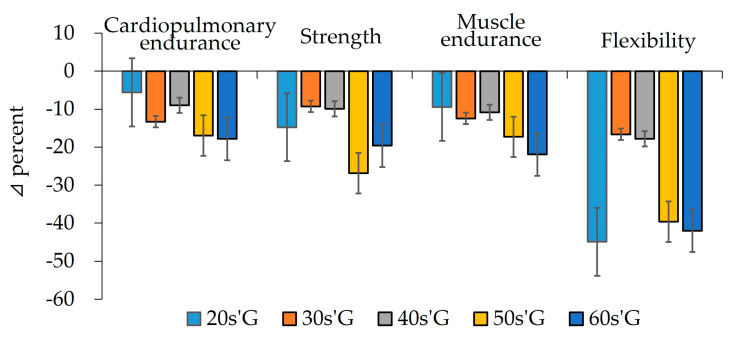
Delta % in physical fitness factors among groups after detraining.

**Figure 4 ijerph-19-01845-f004:**
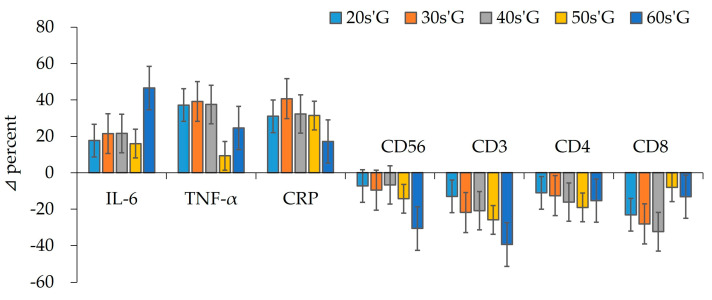
Delta % in cytokines, CRP, and immunocytes among groups after detraining.

**Table 1 ijerph-19-01845-t001:** Physical characteristics of the participants before COVID-19.

	Groups						
	20s’G (*n* = 14; 22%)	30s’G (*n* = 12; 19%)	40s’G (*n* = 12; 19%)	50s’G (*n* = 12; 19%)	60s’G (*n* = 14; 22%)	*X^2^*	*p*
Age (y)	23.29 ± 1.33	35.09 ± 2.07	44.50 ± 2.28	54.92 ± 2.15	63.64 ± 1.95	59.567	0.001
Height (cm)	174.74 ± 3.93	169.62 ± 3.92	171.80 ± 4.51	176.55 ± 4.05	166.36 ± 4.21	29.344	0.001
Weight (kg)	74.44 ± 6.97	73.05 ± 5.51	74.03 ± 9.12	78.28 ± 6.82	65.29 ± 8.14	16.918	0.002
Percent fat (%)	18.34 ± 5.51	17.97 ± 3.87	19.37 ± 4.70	18.54 ± 6.58	19.40 ± 7.28	1.258	0.869
WHR	0.84 ± 0.03	0.84 ± 0.02	0.87 ± 0.04	0.85 ± 0.04	0.84 ± 0.04	7.458	0.114
Detrained periods (wk)	45.83 ± 6.32	42.52 ± 5.72	45.93 ± 5.77	44.21 ± 5.24	46.25 ± 5.19	4.263	0.372

All data represents mean ± standard deviation. Differences between groups were analyzed by the Kruskal–Wallis test. 20s’G, 20s group; 30s’G, 30s group; 40s’G, 40s group; 50s’G, 50s group; 60s’G, 60s group; WHR, waist/hip ratio.

**Table 2 ijerph-19-01845-t002:** Differences and changes in body composition among 5 groups.

		20s’G	30s’G	40s’G	50s’G	60s’G	F	η^2^
Muscle mass	pre	38.60 ± 7.75	47.68 ± 8.86	47.43 ± 9.27	33.99 ± 8.66	33.07 ± 10.02	5.545 ***	0.587
(kg)	post	35.51 ± 7.48	42.23 ± 8.71	42.84 ± 9.73	29.44 ± 6.18	24.99 ± 6.69	6.744 ***	0.683
	Z	−1.915	−2.848 **	−2.551 *	−1.969 *	−3.235 ***		
Fat mass	pre	13.71 ± 4.50	13.06 ± 2.69	14.41 ± 4.03	14.65 ± 5.80	12.41 ± 3.85	0.623	0.199
(kg)	post	16.98 ± 3.17	14.85 ± 3.27	18.76 ± 6.10	18.24 ± 5.69	16.58 ± 5.40	1.413	0.269
	Z	−2.731 **	−1.913	−2.510 *	−2.668 **	−3.297 ***		
Percent fat	pre	18.34 ± 5.51	17.97 ± 3.87	19.37 ± 4.70	18.54 ± 6.58	19.40 ± 7.28	0.471	0.101
(%)	post	21.68 ± 4.22	20.05 ± 4.21	24.58 ± 5.65	24.16 ± 5.55	25.83 ± 8.66	1.082	0.340
	Z	−2.166 *	−1.423	−2.432 *	−2.667 **	−2.982 **		
WHR	pre	0.84 ± 0.03	0.84 ± 0.02	0.87 ± 0.04	0.85 ± 0.04	0.84 ± 0.04	2.166	0.323
	post	0.90 ± 0.03	0.92 ± 0.06	0.95 ± 0.11	0.94 ± 0.10	0.97 ± 0.17	0.843	0.248
	Z	−3.118 **	−2.938 **	−2.654 **	−2.867 **	−2.937 **		

All data represents mean ± standard deviation. 20s’G, 20s group; 30s’G, 30s group; 40s’G, 40s group; 50s’G, 50s group; 60s’G, 60s group; WHR, waist/hip ratio. *, *p* < 0.05; **, *p* < 0.01; ***, *p* < 0.001.

**Table 3 ijerph-19-01845-t003:** Differences and changes in physical fitness among 5 groups.

		20s’G	30s’G	40s’G	50s’G	60s’G	F	η^2^
VO_2_max	pre	61.37 ± 8.13	62.18 ± 7.59	55.91 ± 10.61	52.52 ± 5.80	46.87 ± 7.25	0.836	0.603
(ml/kg/min)	post	57.94 ± 6.51	53.91 ± 5.45	50.89 ± 6.83	43.62 ± 3.33	38.51 ± 6.02	0.519	0.793
	Z	−2.040 *	−2.756 **	−2.118 *	−2.667 **	−3.296 ***		
Grip strength	pre	42.08 ± 9.35	43.94 ± 11.68	39.40 ± 11.32	40.39 ± 7.49	34.16 ± 4.30	1.513	0.360
(kg)	post	35.86 ± 7.95	39.86 ± 11.32	35.48 ± 10.63	29.55 ± 3.15	27.46 ± 4.13	0.659	0.272
	Z	−3.045 **	−2.580 **	−2.432 *	−2.904 **	−3.235 ***		
Sit-ups	pre	59.14 ± 8.21	57.73 ± 8.57	50.75 ± 7.77	50.58 ± 8.67	49.14 ± 7.05	0.785	0.477
(reps)	post	53.57 ± 7.47	50.55 ± 10.20	45.25 ± 9.18	41.83 ± 8.75	38.36 ± 6.89	0.445	0.576
	Z	−2.737 **	−2.849 **	−1.779	−2.670 **	−3.238 ***		
Sit & reach	pre	15.86 ± 6.32	13.72 ± 5.48	13.99 ± 7.67	12.88 ± 7.16	10.24 ± 4.26	0.327	0.305
(cm)	post	8.74 ± 4.26	11.44 ± 5.28	11.50 ± 6.51	7.78 ± 4.58	5.94 ± 3.24	1.934	0.420
	Z	−2.856 **	−0.889	−1.883	−2.589 **	−3.109 **		

All data represents mean ± standard deviation. 20s’G, 20s group; 30s’G, 30s group; 40s’G, 40s group; 50s’G, 50s group; 60s’G, 60s group. *, *p* < 0.05; **, *p* < 0.01; ***, *p* < 0.001.

**Table 4 ijerph-19-01845-t004:** Differences and changes in immunocytes among 5 groups.

		20s’G	30s’G	40s’G	50s’G	60s’G	F	η2
Leucocytes	pre	5.67 ± 11.89	5.93 ± 10.82	5.41 ± 8.11	5.71 ± 6.67	5.88 ± 6.95	0.814	0.201
(×103/μL)	post	5.96 ± 9.66	6.26 ± 10.16	6.02 ± 8.85	6.20 ± 7.45	6.35 ± 6.38	0.545	0.181
	Z	−1.262	−1.511	−1.608	−1.571	−2.198 *		
NK cell	pre	30.72 ± 6.18	30.58 ± 7.33	28.53 ± 4.87	25.09 ± 7.57	18.55 ± 8.69	1.401	0.569
(%)	post	28.48 ± 7.98	27.65 ± 6.65	26.61 ± 6.30	21.51 ± 6.97	12.87 ± 5.43	3.191*	0.682
	Z	−1.100	−1.481	−1.067	−1.680	−2.497 *		
CD3	pre	66.36 ± 8.91	67.09 ± 10.36	64.69 ± 19.67	64.71 ± 10.07	55.66 ± 16.42	0.528	0.308
(%)	post	57.74 ± 10.11	52.42 ± 16.14	51.22 ± 14.46	47.94 ± 18.06	33.74 ± 14.87	0.837	0.512
	Z	−2.623 **	−2.524 *	−2.904 **	−2.666 **	−3.181 ***		
CD4	pre	44.36 ± 6.00	42.81 ± 5.27	43.17 ± 9.19	40.03 ± 9.51	34.23 ± 11.03	0.514	0.418
(%)	post	39.46 ± 7.22	37.42 ± 7.01	36.22 ± 11.29	32.40 ± 8.74	28.99 ± 8.65	0.508	0.421
	Z	−2.299 *	−2.032 *	−1.826	−2.320 *	−2.032 *		
CD8	pre	35.65 ± 7.31	34.78 ± 8.52	30.88 ± 8.97	20.86 ± 6.70	18.14 ± 3.81	1.484	0.729
(%)	post	27.41 ± 7.73	25.01 ± 7.07	20.88 ± 6.75	19.19 ± 4.64	15.75 ± 3.44	0.583	0.589
	Z	−2.570 **	−2.386 *	−2.070 *	−1.625	−1.663		

All data represents mean ± standard deviation. 20s’G, 20s group; 30s’G, 30s group; 40s’G, 40s group; 50s’G, 50s group; 60s’G, 60s group; NK, natural killer; CD, cluster of differentiation. *, *p* < 0.05; **, *p* < 0.01; ***, *p* < 0.001.

**Table 5 ijerph-19-01845-t005:** Differences and changes in cytokines among 5 groups.

		20s’G	30s’G	40s’G	50s’G	60s’G	F	η2
IL-6	pre	11.58 ± 4.55	11.45 ± 4.53	15.85 ± 4.21	18.85 ± 4.97	17.77 ± 3.67	2.895 *	0.590
(pg/mL)	post	13.63 ± 5.02	13.91 ± 4.71	19.27 ± 3.82	21.87 ± 5.40	26.04 ± 6.63	0.954	0.693
	Z	−2.040 *	−2.134 *	−2.197 *	−1.257	−2.982 **		
TNF-α	pre	22.68 ± 10.09	21.01 ± 5.46	22.32 ± 5.24	35.92 ± 9.41	32.15 ± 6.83	4.459 **	0.625
(pg/mL)	post	31.11 ± 12.29	29.24 ± 7.37	30.69 ± 5.30	39.26 ± 9.29	40.06 ± 12.00	0.963	0.438
	Z	−2.605 **	−2.667 **	−2.824 **	−0.393	−2.103 *		
CRP	pre	26.24 ± 8.81	29.59 ± 6.60	26.17 ± 5.06	36.14 ± 11.44	44.86 ± 13.17	2.576 *	0.621
(pg/mL)	post	34.38 ± 9.78	41.63 ± 6.01	34.62 ± 6.89	47.50 ± 10.33	52.58 ± 13.84	6.067 ***	0.608
	Z	−2.668 **	−2.934 **	−3.059 **	−2.353 *	−2.166 *		

All data represents mean ± standard deviation. 20s’G, 20s group; 30s’G, 30s group; 40s’G, 40s group; 50s’G, 50s group; 60s’G, 60s group; IL, interleukin; TNF, tumor necrosis factor; CRP, C-reactive protein. *, *p* < 0.05; **, *p* < 0.01; ***, *p* < 0.001.

## Data Availability

The data presented in this study are available on request from the corresponding authors. The data are not publicly available because this study is not yet finished.

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
