# Peer review of "Detraining Effects of COVID-19 Pandemic on Physical Fitness, Cytokines, C-Reactive Protein and Immunocytes in Men of Various Age Groups"

_ijerph, 2022, doi:10.3390/ijerph19031845_

Round 1

Reviewer 1 Report

The present paper was conducted in order to find differences in physical fitness, immunocytes and cytokines before and during the COVID-19 pandemic.

The manuscript addresses scientifically and clinically relevant questions, despite this some concerns should be taken into consideration.

Aims (Primary, Secondary, ….) should be stated clearly both in the abstract and at the end of the introduction. At the end of Introduction aims are very confused and presents many repetitions (lines 63-70).

Authors should clarify why they only measured TNF-alpha and IL-6 among the pro-inflammatory cytokines. In addition, CRP is not a cytokine, but a plasma protein.

In Materials and Methods section, authors should better describe enrolled subjects and type of exercise, it’s not sufficient to say “30-100 min/day, 3-5 days/week for over 6 months before the COVID-19 pandemic” and thus discuss accordingly. Please refers also to “The WHO Guidelines on physical activity and sedentary behavior, 2020”.

Author should consider to add subheadings when describing “Physical fitness measure”

“Cytokines measures” should be described with more details. What instrument was used?

In results section authors shouldn’t discuss results (lines 225-227; lines 238-240; lines 251-253;lines 266-268;

In discussion section authors should’t insert new results, and move them in the results section (Figures 2, 3, 4). In addition more discussion on “Physical fitness measure” should be added.

In conclusion, the article should be entirely corrected by a native speaking English; the manuscript needs careful editing to improve the flow of information, correct word usage, and repetitions.

Author Response

Answers to 1st reviewer’s comments

Thank you for your kind advice and comments about this manuscript. We revised our manuscript as per your comments. We represented the specific modifications in response to the comments by blue-letters in our manuscript. We sincerely appreciate your comments because your comments make our manuscript better. Details of responses about reviewer’s comments are as follows:

#1. Comments or Suggestions: Aims (Primary, Secondary, ….) should be stated clearly both in the abstract and at the end of the introduction. At the end of Introduction aims are very confused and presents many repetitions (lines 63-70).

Response: Thank you for your comments. Based on your comment, we have supplemented in the abstract (lines 15-16 of revised new version) as follows: “The aim of this study was to investigate the detraining effects of the COVID-19 confinement period on physical fitness, immunocytes, inflammatory cytokines, and proteins in various age groups.” In addition, we revised at the end of introduction (lines 69-73 of revised new version) as follows: “In addition, studies that focus on various age groups are lacking. Therefore, the aim of this study was to investigate the detraining effects of the COVID-19 pandemic on immunocytes, CRP, and cytokines in various age groups, as well as their effects on physical fitness related to the above variables.”

#2. Comments or Suggestions: Authors should clarify why they only measured TNF-alpha and IL-6 among the pro-inflammatory cytokines. In addition, CRP is not a cytokine, but a plasma protein.

Response: Thank you for your comments. We wanted to explain "chronic inflammation is reflected by increased C-reactive protein (CRP) concentrations and increased systemic levels of some cytokines". In this process, an error occurred that CRP could be bound to cytokines. All contents of the title, abstract and text have been revised according to your comments. The title has been modified as follows. “Detraining Effects of COVID-19 Pandemic on Physical Fitness, Cytokines, C-reactive Protein, and Immunocytes in Men of Various Age Groups” In this regard, the following sentences were added between lines 49-51 (new version) and references were also added. “…Moreover, chronic inflammatory conditions increase C-reactive protein (CRP) concentrations as well as some cytokines such as interleukin (IL)-6 and tumor necrosis factor (TNF)-α, leading to immune-related diseases [8].”

[8] Petersen, A.M.W.; Pedersen, B.K. The anti-inflammatory effect of exercise. J. Appl. Physiol. 2005, 98, 1154–1162. doi:10.1152/japplphysiol.00164.2004.

#3. Comments or Suggestions: In Materials and Methods section, authors should better describe enrolled subjects and type of exercise, it’s not sufficient to say “30-100 min/day, 3-5 days/week for over 6 months before the COVID-19 pandemic” and thus discuss accordingly. Please refers also to “The WHO Guidelines on physical activity and sedentary behavior, 2020”.

Response: Thank you for your good comments and references. However, the reviewer pointed out that the subjects of this study were 'the explanation of how exercise was performed before COVID-19'. If you look at the previous sentence, 'The average (± SD) period during which they did not enter the sports facility was 44.85 (± 5.65) weeks, ....' These sentences seem to be confusing, so we have modified them as follows. ”The average (± SD) period during which the subjects in this study did not use sports facilities and did not participate in regular exercise was 44.85 (± 5.65) weeks, although they had exercised for 30-100 min/day, 3-5 days/week for over 6 months prior to the COVID-19 pandemic…” on new revised version lines 79-82. In addition, the detrained period due to inability to use sports facilities for each age group is presented in Table 1.

#4. Comments or Suggestions: Author should consider to add subheadings when describing “Physical fitness measure”

Response: Thank you for your good comments. Based on your comments, we have added subheadings on new revised version lines 114, 125, 131, 139, and 147, respectively.

#5. Comments or Suggestions: “Cytokines measures” should be described with more details. What instrument was used?

Response: Thank you for your good comments. Based on your comments, we have added sentences on new revised version lines 206-213 as follows. “The CRP test was carried out using the Immunoturbidity-based CardioPhase hsCRP (Dade Behring, Marburg, Germany) reagent mounted on the BNII (Dade Behring, Marburg, Germany) equipment. The full-scale CRP test using the immunoturbidity method was performed using Nanopia CRP (N-CRP; Sekisui, Tokyo, Japan) and IATRO CRP-EX (I-CRP; Iatron, Tokyo, Japan). CRP reagents with PureAuto S CRP-SS type (P-CRP; Sekisui) were installed on a Hitachi 7600-110 automated analyzer (Hitachi High-Technologies Co., Tokyo, Japan), and the dilution factor of the equipment was not changed.”

#6. Comments or Suggestions: In results section authors shouldn’t discuss results (lines 225-227; lines 238-240; lines 251-253; lines 266-268)

Response: Thank you for your good comments. The part you pointed out has been deleted or moved to the 'Discussion' section as follows.

Lines 225-227: This is the result of the decrease in the amount of exercise that has been done consistently in all age groups, suggesting that the detraining can have harmful effects on the human body. In particular, it can be said that it leads to worse outcomes in the elderly.

Lines 238-240: These results indicate that the COVID-19 period negatively affected health-related physical fitness components in the various age groups.

Lines 251-253: These changes suggest that the function of immune cells is generally lowered by physical inactivity, although these reduced levels are not abnormal.

Lines 266-268: The meaning of these results can be interpreted as implying that physical inactivity can increase the inflammatory state in all age groups.

#7. Comments or Suggestions: In discussion section authors should’t insert new results, and move them in the results section (Figures 2, 3, 4). In addition more discussion on “Physical fitness measure” should be added.

Response: Thank you for your good comments. All result data from the discussion part was moved to 'Results' and explanations were also added. Specifically, the revised contents are as follows.

New revised version lines 249-255:

“As shown in Fig. 2, after detraining due to COVID-19, the muscle mass in 20s'G decreased by about 8%, 30s'G by 11%, 40s'G by 10%, 50s'G by 13%, and 60s'G by 24%. After detraining, fat mass increased by about 24% in 20s'G, 14% in 30s'G, 30% in 40s'G, 25% in 50s'G, and 34% in 60s'G. Similar to fat mass, fat percentage increased by about 18% in 20s'G, 12% in 30s'G, 27% in 40s'G, 30% in 50s'G, and 33% in 60s'G. WHR showed a steady rise with increasing age. That is, in the case of 20s'G, WHR increased by about 7%, 30s'G by 10%, 40s'G by 9%, 50s'G by 11%, and 60s'G by 15%.”

New revised version lines 271-277:

“As shown in Fig. 3, after detraining due to COVID-19 confinement, the cardiopulmonary endurance in 20s'G decreased by about 6%, 30s'G by 13%, 40s'G by 9%, 50s'G by 17%, and 60s'G by 18%. After detraining, strength decreased by about 15% in 20s'G, 9% in 30s'G, 10% in 40s'G, 27% in 50s'G, and 20% in 60s'G. Similar to strength, muscle endurance decreased by about 9% in 20s'G, 12% in 30s'G, 11% in 40s'G, 17% in 50s'G, and 22% in 60s'G. Flexibility also decreased after detraining. In 20s'G, it reduced by about 45%, 30s'G by 17%, 40s'G by 18%, 50s'G by 40%, and 60s'G by 42%.”

New revised version lines 306-314:

“As shown in Fig. 4, after detraining due to COVID-19, the IL-6 in 20s'G increased by about 18%, 30s'G by 21%, 40s'G by 22%, 50s'G by 16%, and 60s'G by 47%. After detraining, TNF-α increased by about 37% in 20s'G, 39% in 30s'G, 38% in 40s'G, 9% in 50s'G, and 25% in 60s'G. CRP also increased by about 31% in 20s'G, 41% in 30s'G, 32% in 40s'G, 31% in 50s'G, and 17% in 60s'G. Meanwhile, CD56 decreased by about 7% in 20s'G, 10% in 30s'G, 7% in 40s'G, 14% in 50s'G, and 31% in 60s'G. CD3 decreased by about 13% in 20s'G, 22% in 30s'G, 21% in 40s'G, 26% in 50s'G, and 39% in 60s'G. CD4 decreased by about 11% in 20s'G, 13% in 30s'G, 16% in 40s'G, 19% in 50s'G, and 15% in 60s'G. At last, CD8 decreased by about 23% in 20s'G, 28% in 30s'G, 32% in 40s'G, 8% in 50s'G, and 13% in 60s'G.”

Once again, thanks for your review.

Reviewer 2 Report

The manuscript is of high quality and confirms what was expected from post-pandemic confinement. However, despite the good design of the study and the detail expressed by the authors, there is a major limitation in the participants, who are only male. In this sense, it is recommended that these characteristics be specified in the title, as this may lead to confusion. It is also recommended that this information be included in the abstract. 
In the abstract, "introduction and objective" is stated, but the objective is not specified. In addition, the total number of participants and gender are not stated. Express the meaning of the acronym COVID-19 the first time it appears. 
Section 2.1. "participants" includes information about the procedure, which should be included in another section. In this respect, the procedure should state when the first measurement was carried out, before the pandemic, but when? 
I find that the groups are small in number and perhaps it would be interesting to propose a group that includes more people and a wider age range. Since the physiological differences between a person of 20 and a person of 30 are infinite. Perhaps the results of the study could be improved by combining several groups. 
Table 1 should be included in the results, as it shows more variables than just the age of the participants. Furthermore, the table should include the N and % that each group represents with respect to the total sample, e.g. add a variable in column 1 that is n (%). 
In section 2.2.1. it is recommended to put the reliability of the assessment tests carried out. 
Figure 1 is of poor quality, perhaps it would be better to enlarge it. 
Line 202 and 203 repeats information about the participants, it is not relevant in this section.
It is recommended to merge table 1 and 2 and denote something like "physical and body composition characteristics of the sample...".
The limitations should be included in the discussion section and not in the conclusions. It is therefore recommended to delete this part and specify the conclusions based on the stated objectives. 
In addition, the lines for the future and the practical applications of the study are missing. 

Author Response

Answers to 2nd reviewer’s comments

Thank you for your kind advice and comments about this manuscript. We revised our manuscript as per your comments. We represented the specific modifications in response to the comments by blue-letters in our manuscript. We sincerely appreciate your comments because your comments make our manuscript better. Details of responses about reviewer’s comments are as follows:

#1. Comments or Suggestions: The manuscript is of high quality and confirms what was expected from post-pandemic confinement. However, despite the good design of the study and the detail expressed by the authors, there is a major limitation in the participants, who are only male. In this sense, it is recommended that these characteristics be specified in the title, as this may lead to confusion. It is also recommended that this information be included in the abstract.

Response: First of all, thank you for comments. Based on your comments, we have revised in the title as follows: “Detraining Effects of COVID-19 Pandemic on Physical Fitness, Cytokines, C-reactive Protein, and Immunocytes in Men of Various Age Groups”, and in the abstract as follows: “The participants of this study included sixty-four male adults who did not exercise during the pandemic, although they had exercised regularly before.”

#2. Comments or Suggestions: In the abstract, "introduction and objective" is stated, but the objective is not specified. In addition, the total number of participants and gender are not stated. Express the meaning of the acronym COVID-19 the first time it appears.

Response: Based on your comments, we have revised in the abstract as follows:

(1) Abstract part on lines 12-19: “Since the start of the COVID-19 pandemic caused by severe acute respiratory syndrome coronavirus II, levels of physical inactivity have become more severe and widespread than ever before. Physical inactivity is known to have a negative effect on the human body, but the extent to which reduced physical fitness has on immune function before and after the current pandemic has yet to be found. The aim of this study was to investigate the detraining effects of the COVID-19 confinement period on physical fitness, immunocytes, inflammatory cytokines, and proteins in various age groups. The participants of this study included sixty-four male adults who did not exercise during the pandemic, although they had exercised regularly before.”

(2) Introduction part on lines 67-73: “There has been much interest regarding the effects of physical fitness on immunocytes, CRP, and cytokines, but research related to reduced physical training or fitness on those variables has not been conducted until now. In addition, studies that focus on various age groups are lacking. Therefore, the aim of this study was to investigate the detraining effects of the COVID-19 pandemic on immunocytes, CRP, and cytokines in various age groups, as well as their effects on physical fitness related to the above variables.”

#3. Comments or Suggestions: Section 2.1. "participants" includes information about the procedure, which should be included in another section. In this respect, the procedure should state when the first measurement was carried out, before the pandemic, but when?

Response: Thank you for your comments. According to your comments, we revised the date as follows.

On new revised version lines 77-79: “…. Male participants who could not exercise regularly from March 9, 2020 to March 28, 2021 due to COVID-19 social distancing rules were recruited.”

#4. Comments or Suggestions: I find that the groups are small in number and perhaps it would be interesting to propose a group that includes more people and a wider age range. Since the physiological differences between a person of 20 and a person of 30 are infinite. Perhaps the results of the study could be improved by combining several groups.

Response: Your opinion is right. We tried to recruit more participants during the research plan, but it is too difficult to recruit and control participants due to COVID-19. Further studies would like to investigate the untrained effects of physical activity or exercise in a larger number of participants. Please understand it in a broad sense. Nevertheless, to fit the statistical procedure, we used the G*power program to fit the number of subjects. The contents are as follows. “….G*Power 3.1.9.7 was used to calculate the sample with an effect size of 0.40, a significance level of 0.05, and a power of 0.80 required for the ANCOVA test.”

#5. Comments or Suggestions: Table 1 should be included in the results, as it shows more variables than just the age of the participants. Furthermore, the table should include the N and % that each group represents with respect to the total sample, e.g. add a variable in column 1 that is n (%).

Response: Thank you for your comments. According to your comments, we inserted the n (%) in the Table 1.  In addition to age, in Table 1, height, weight, percentage fat, WHR, and detrained periods were inserted. For reference, Table 1 showing the study subjects in most studies is included in the 'Materials and Methods', so it is placed on lines 106-108. I hope you understand.

#6. Comments or Suggestions: In section 2.2.1. it is recommended to put the reliability of the assessment tests carried out.

Response: Based on your comments, we calculated the reliability and included the Cronbach's α in the text as follows. On line 127, “Cronbach's α indicating the reliability of the flexibility test was 0.725.” On line 138, “Cronbach's α indicating the reliability of the strength test was 0.831.” On line 146, “Cronbach's α indicating the reliability of the muscle endurance test was 0.827.” On line 167, “Cronbach's α indicating the reliability of the cardiopulmonary endurance test was 0.843.”

#7. Comments or Suggestions: Figure 1 is of poor quality, perhaps it would be better to enlarge it.

Response: Based on your comments, we revised it as follows.

#8. Comments or Suggestions: Line 202 and 203 repeats information about the participants, it is not relevant in this section.

Response: Based on your comments, we have deleted.

#9. Comments or Suggestions: It is recommended to merge table 1 and 2 and denote something like "physical and body composition characteristics of the sample...".

Response: Table 1 and Table 2 provide different information. Merging the two would make it difficult to provide understanding to readers. Please understand.

#10. Comments or Suggestions: The limitations should be included in the discussion section and not in the conclusions. It is therefore recommended to delete this part and specify the conclusions based on the stated objectives.

Response: Based on your comments, we have deleted and moved the limitations in the discussion section.

#11. Comments or Suggestions: In addition, the lines for the future and the practical applications of the study are missing.

Response: According to your suggestion, we inserted below sentences in the lines 485-487 as follows. “Therefore, a strategy that promotes regular physical exercise is needed, even before COVID-19 vaccines and therapies are developed and distributed.”

Once again, thank you very much for your review.
